# DIVIDE-AND-CONQUER MONTE CARLO TREE SEARCH

## ABSTRACT

Standard planners for sequential decision making (including Monte Carlo planning, tree search, dynamic programming, etc.) are constrained by an implicit *sequential planning assumption*: The order in which a plan is constructed is the same in which it is executed. We consider alternatives to this assumption for the class of goal-directed Reinforcement Learning (RL) problems. Instead of an environment transition model, we assume an imperfect, goal-directed policy. This low-level policy can be improved by a plan, consisting of an appropriate sequence of sub-goals that guide it from the start to the goal state. We propose a planning algorithm, Divide-and-Conquer Monte Carlo Tree Search (DC-MCTS), for approximating the optimal plan by means of proposing intermediate sub-goals which hierarchically partition the initial tasks into simpler ones that are then solved independently and recursively. The algorithm critically makes use of a learned sub-goal proposal for finding appropriate partition trees of new tasks based on prior experience. Different strategies for learning sub-goal proposals give rise to different planning strategies that strictly generalize sequential planning. We show that this algorithmic flexibility wrt. planning order leads to improved results in navigation tasks in grid-worlds as well as in challenging continuous control environments.

## 1 INTRODUCTION

This is the first sentence of this paper, but it was not the first one we wrote. In fact, the entire introduction section was actually one of the last sections to be added to this manuscript. The discrepancy between the order of inception of ideas and the order of their presentation in this paper probably does not come as a surprise to the reader. Nonetheless, it serves as a point for reflection that is central to the rest of this work, and that can be summarized as *"the order in which we construct a plan does not have to coincide with the order in which we execute it"*.

Most standard planners for sequential decision making problems—including Monte Carlo planning, Monte Carlo Tree Search (MCTS) and dynamic programming—have a baked-in *sequential planning assumption* (Bertsekas et al., 1995; Browne et al., 2012). These methods begin at either the initial or final state and then plan actions sequentially forward or backwards in time. However, this sequential approach faces two main challenges. (i) The transition model used for planning needs to be reliable over long horizons, which is often difficult to achieve when it has to be inferred from data. (ii) Credit assignment to each individual action is difficult: In a planning problem spanning a horizon of 100 steps, to assign credit to the first action, we have to compute the optimal cost-to-go for the remaining problem with a horizon of 99 steps, which is only slightly easier than solving the original problem.

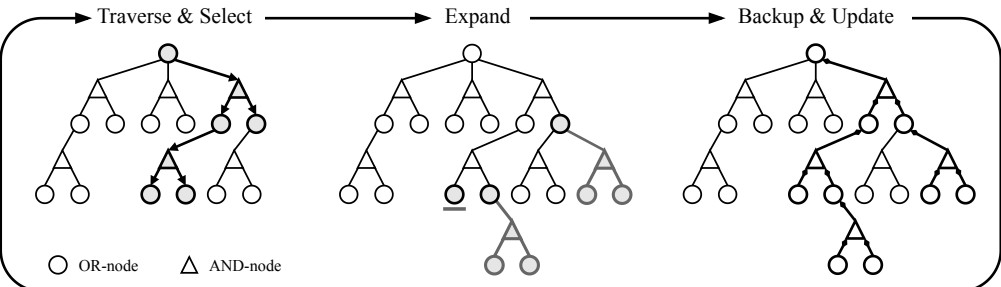

Figure 1: Divide-and-Conquer Monte Carlo Tree Search (DC-MCTS).

To overcome these two fundamental challenges, here we consider alternatives to the basic assumptions of sequential planners. We focus on goal-directed decision making problems where an agent should reach a goal state from a start state. Instead of a transition and reward model of the environment, we assume a given goal-directed policy (the "low-level" policy) and the associated value oracle that returns its success probability on any given task.[1] In general, a low-level policy will not be not optimal, e.g. it might be too "myopic" to reliably reach goal states that are far away from its current state. We now seek to improve the low-level policy via a suitable *sequence of sub-goals* that guide it from the start to the final goal, thus maximizing the overall task success probability. This formulation of planning as finding good sub-goal sequences, makes learning of explicit environment models unnecessary, as they are replaced by low-level policies and their value functions.

The sub-goal planning problem can still be solved by a conventional sequential planner that begins by searching for the first sub-goal to reach from the start state, then planning the next sub-goal in sequence, and so on. Indeed, this is the approach taken in most hierarchical RL settings based on options or sub-goals (e.g. Dayan & Hinton, 1993; Sutton et al., 1999; Vezhnevets et al., 2017). However, the credit assignment problem mentioned above persists, as assessing if the first sub-goal is useful still requires evaluating the success probability of the remaining plan. Instead, it could be substantially easier to reason about the utility of a sub-goal "in the middle" of the plan, as this breaks the long-horizon problem into two sub-problems with much shorter horizons: how to get to the sub-goal and how to get from there to the final goal. Based on this intuition, we propose the Divide-and-Conquer MCTS (DC-MCTS) planner that searches for sub-goals to split the original task into two independent sub-tasks of comparable complexity and then recursively solves these, thereby drastically facilitating credit assignment. To search the space of intermediate sub-goals efficiently, DC-MCTS uses a heuristic for proposing promising sub-goals that is learned from previous search results and agent experience.

Humans can plan efficiently over long horizons to solve complex tasks, such as theorem proving or navigation, and some plans even span over decades (e.g. economic measures): In these situations, planning sequentially in terms of next steps – such as what arm to move, or what phone call to make – will cover a tiny proportion of the horizon, neglecting the long uncertainty beyond the last planned step. The algorithm put forward in this paper is a step in the direction of efficient planners that tackle long horizons by recursively and parallelly splitting them into many smaller and smaller sub-problems. In Section 2, we formulate planning in terms of sub-goals instead of primitive actions. In Section 3, as our main contribution, we propose the novel Divide-and-Conquer Monte Carlo Tree Search algorithm for this planning problem. In Section 4 we position DC-MCTS within the literature of related work. In Section 5, we show that it outperforms sequential planners both on grid world and continuous control navigation tasks, demonstrating the utility of constructing plans in a flexible order that can be different from their execution order.

## 2 IMPROVING GOAL-DIRECTED POLICIES WITH PLANNING

Let $\mathcal{S}$ and $\mathcal{A}$ be finite sets of states and actions. We consider a multi-task setting, where for each episode the agent has to solve a new task consisting of a new Markov Decision Process (MDP) $\mathcal{M}$ over $\mathcal{S}$ and $\mathcal{A}$. Each $\mathcal{M}$ has a single start state $s_0$ and a special absorbing state $s_\infty$, also termed the goal state. If the agent transitions into $s_\infty$ at any time it receives a reward of 1 and the episode terminates; otherwise the reward is 0. We assume that the agent observes the start and goal states $(s_0, s_\infty)$ at the beginning of each episode, as well as an encoding vector $c_\mathcal{M} \in \mathbb{R}^d$. This vector provides the agent with additional information about the MDP $\mathcal{M}$ of the current episode and will be key to transfer learning across tasks in the multi-task setting. A stochastic, goal-directed policy $\pi$ is a mapping from $\mathcal{S} \times \mathcal{S} \times \mathbb{R}^d$ into distributions over $\mathcal{A}$, where $\pi(a|s, s_\infty, c_\mathcal{M})$ denotes the probability of taking action $a$ in state $s$ in order to get to goal $s_\infty$. For a fixed goal $s_\infty$, we can interpret $\pi$ as a regular policy, here denoted as $\pi_{s_\infty}$, mapping states to action probabilities. We denote the value of $\pi$ in state $s$ for goal $s_\infty$ as $v^\pi(s, s_\infty | c_\mathcal{M})$; we assume no discounting $\gamma = 1$. Under the above definition of the reward, the value is equal to the success probability of $\pi$ on the task, i.e. the absorption probability of the stochastic process starting in $s_0$ defined by running $\pi_{s_\infty}$:

$$v^\pi(s_0, s_\infty | c_\mathcal{M}) = P(s_\infty \in \tau_{s_0}^{\pi_{s_\infty}} | c_\mathcal{M}),$$

---

[1] As we will observe in Section 5, in practice both the low-level policy and value can be learned. Approximating the value oracle with a learned value function was sufficient for DC-MCTS to plan successfully.

where $\tau_{s_0}^{\pi_{s_\infty}}$ is the trajectory generated by running $\pi_{s_\infty}$ from state $s_0$ [2]. To keep the notation compact, we will omit the explicit dependence on $c_\mathcal{M}$ and abbreviate tasks with pairs of states in $\mathcal{S} \times \mathcal{S}$.

## 2.1 PLANNING OVER SUB-GOAL SEQUENCES

Assume a given goal-directed policy $\pi$, which we also refer to as the low-level policy. If $\pi$ is not already optimal, we can potentially improve it by planning: If $\pi$ has a low probability of directly reaching $s_\infty$ from the initial state $s_0$, i.e. $v^\pi(s_0, s_\infty) \approx 0$, we will try to find a *plan* consisting of a sequence of intermediate sub-goals such that they guide $\pi$ from the start $s_0$ to the goal state $s_\infty$.

Concretely, let $\mathcal{S}^* = \cup_{n=0}^\infty \mathcal{S}^n$ be the set of sequences over $\mathcal{S}$, and let $|\sigma|$ be the length of a sequence $\sigma \in \mathcal{S}^*$. We define for convenience $\bar{\mathcal{S}} := \mathcal{S} \cup \{\varnothing\}$, where $\varnothing$ is the empty sequence representing no sub-goal. We refer to $\sigma$ as a *plan* for task $(s_0, s_\infty)$ if $\sigma_1 = s_0$ and $\sigma_{|\sigma|} = s_\infty$, i.e. if the first and last elements of $\sigma$ are equal to $s_0$ and $s_\infty$, respectively. $s_0 \mathcal{S}^* s_\infty$ denotes the set of plans for this task.

To execute a plan $\sigma$, we construct a policy $\pi_\sigma$ by conditioning the low-level policy $\pi$ on each of the sub-goals in order: Starting with $n = 1$, we feed sub-goal $\sigma_{n+1}$ to $\pi$, i.e. we run $\pi_{\sigma_{n+1}}$; if $\sigma_{n+1}$ is reached, we will execute $\pi_{\sigma_{n+2}}$ and so on. We now wish to do *open-loop planning*, i.e. find the plan with the highest success probability $P(s_\infty \in \tau_{s_0}^{\pi_\sigma})$ of reaching $s_\infty$. However, this success probability depends on the transition kernels of the underlying MDPs, which might not be known. We can instead define planning as maximizing the following lower bound of the success probability, that can be expressed in terms of the low-level value $v^\pi$.

**Proposition 1** (Lower bound of success probability). *The success probability $P(s_\infty \in \tau_{s_0}^{\pi_\sigma}) \geq L(\sigma)$ of a plan $\sigma$ is bounded from below by $L(\sigma) := \prod_{i=1}^{|\sigma|-1} v^\pi(\sigma_i, \sigma_{i+1})$, i.e. the product of the success probabilities of $\pi$ on the sub-tasks defined by $(\sigma_i, \sigma_{i+1})$.*

The straight-forward proof is given in Appendix A.1. Intuitively, $L(\sigma)$ is a lower bound for the success of $\pi_\sigma$, as it neglects the probability of "accidentally" (due to stochasticity of the policy or transitions) running into the goal $s_\infty$ before having executed the full plan. We summarize:

**Definition 1** (Open-Loop Goal-Directed Planning). *Given a goal-directed policy $\pi$ and its corresponding value oracle $v^\pi$, we define planning as maximizing $L(\sigma)$ over $\sigma \in s_0 \mathcal{S}^* s_\infty$, i.e. the set of plans for task $(s_0, s_\infty)$. We define the* high-level (HL) value $v^*(s_0, s_\infty) := \max_\sigma L(\sigma)$ *as the maximum value of the planning objective.*

Note the difference between the low-level value $v^\pi$ and the high-level $v^*$. $v^\pi(s, s')$ is the probability of the agent *directly* reaching $s'$ from $s$ following $\pi$, whereas $v^*(s, s')$ the probability reaching $s'$ from $s$ under *the optimal plan*, which likely includes intermediate sub-goals. In particular, $v^* \geq v^\pi$.

## 2.2 AND/OR SEARCH TREE REPRESENTATION

In the following we cast the planning problem into a representation amenable to efficient search. To this end, we use the natural compositionality of plans: We can concatenate a plan $\sigma$ for the task $(s, s')$ and a plan $\hat{\sigma}$ for the task $(s', s'')$ into a plan $\sigma \circ \hat{\sigma}$ for the task $(s, s'')$. Conversely, we can decompose any given plan $\sigma$ for task $(s_0, s_\infty)$ by splitting it at any sub-goal $s \in \sigma$ into $\sigma = \sigma^l \circ \sigma^r$, where $\sigma^l$ is the "left" sub-plan for task $(s_0, s)$, and $\sigma^r$ is the "right" sub-plan for task $(s, s_\infty)$. Trivially, the planning objective and the optimal high-level value factorize wrt. to this decomposition:

$$L(\sigma^l \circ \sigma^r) = L(\sigma^l)L(\sigma^r)$$
$$v^*(s_0, s_\infty) = \max_{s \in \bar{\mathcal{S}}} v^*(s_0, s) \cdot v^*(s, s_\infty).$$

This allows us to recursively reformulate planning as:

$$\arg\max_{s \in \bar{\mathcal{S}}} \left( \arg\max_{\sigma_l \in s_0 \mathcal{S}^* s} L(\sigma^l) \right) \cdot \left( \arg\max_{\sigma_r \in s\mathcal{S}^* s_\infty} L(\sigma^r) \right). \tag{1}$$

The above equations are the Bellman equations and the Bellman optimality equations for the classical single pair shortest path problem in graphs, where edge weights are given by $-\log v^\pi(s, s')$. We can represent this planning problem by an AND/OR search tree (Nilsson, N. J., 1980) with alternating levels of OR and AND nodes. An OR node, also termed an *action node*, is labeled by a task

---

[2]We assume MDPs with multiple absorbing states such that this probability is not trivially equal to 1 for most policies, e.g. uniform policy. In experiments, we used a finite episode length.

$(s, s'') \in \mathcal{S} \times \mathcal{S}$; the root of the search tree is an OR node labeled by the original task $(s_0, s_\infty)$. A terminal OR node $(s, s'')$ has a value $v^\pi(s, s'')$ attached to it, which reflects the success probability of $\pi_{s''}$ for completing the sub-task $(s, s'')$. Each non-terminal OR node has $|\mathcal{S}| + 1$ AND nodes as children. Each of these is labeled by a triple $(s, s', s'')$ for $s' \in \bar{\mathcal{S}}$, which correspond to inserting a sub-goal $s'$ into the overall plan, or not inserting one in case of $s = \varnothing$. Every AND node $(s, s', s'')$, or *conjunction node*, has two OR children, the "left" sub-task $(s, s')$ and the "right" sub-task $(s', s'')$.

In this representation, plans are induced by *solution trees*. A solution tree $\mathcal{T}_\sigma$ is a sub-tree of the complete AND/OR search tree, with the properties that (i) the root $(s_0, s_\infty) \in \mathcal{T}_\sigma$, (ii) each OR node in $\mathcal{T}_\sigma$ has at most one child in $\mathcal{T}_\sigma$ and (iii) each AND node in $\mathcal{T}_\sigma$ as two children in $\mathcal{T}_\sigma$. The plan $\sigma$ and its objective $L(\sigma)$ can be computed from $\mathcal{T}_\sigma$ by a depth-first traversal of $\mathcal{T}_\sigma$. The correspondence of sub-trees to plans is many-to-one, as $\mathcal{T}_\sigma$, in addition to the plan itself, contains *the order* in which the plan was constructed. Figure 6 in Section 5.3 shows an example for a search and solution tree. Below we will discuss how to construct a favourable search order heuristic.

## 3 BEST-FIRST AND/OR PLANNING

The planning problem from Definition 1 can be solved exactly by formulating it as shortest path problem from $s_0$ to $s_\infty$ on a fully connected graph with vertex set $\mathcal{S}$ with non-negative edge weights given by $-\log v^\pi$ and applying a classical Single Source or All Pairs Shortest Path (SSSP / APSP) planner. This approach is appropriate if one wants to solve all goal-directed tasks in a *single* MDP. Here, we focus however on the multi-task setting described above, where the agent is given a new MDP with a single task $(s_0, s_\infty)$ *every* episode. In this case, solving the SSSP / APSP problem is not feasible: Tabulating all graphs weights $-\log v^\pi(s, s')$ would require $|\mathcal{S}|^2$ evaluations of $v^\pi(s, s')$ for all pairs $(s, s')$. In practice, approximate evaluations of $v^\pi$ could be implemented by e.g. actually running the policy $\pi$, or by calls to a powerful function approximator, both of which are often too costly to exhaustively evaluate for large

---

**Algorithm 1** Divide-and-Conquer MCTS

Global low-level value oracle $v^\pi$
Global high-level value function $v$
Global policy prior $p$
Global search tree $\mathcal{T}$

1: **procedure** TRAVERSE(OR node $(s, s'')$)
2:     **if** $(s, s'') \notin \mathcal{T}$ **then**
3:         $\mathcal{T} \leftarrow$ EXPAND$(\mathcal{T}, (s, s''))$
4:         **return** $\max(v^\pi(s, s''), v(s, s''))$   ▷ bootstrap
5:     $s' \leftarrow$ SELECT$(s, s'')$   ▷ OR node
6:     **if** $s' = \varnothing$ **or** max-depth reached **then**
7:         $G \leftarrow v^\pi(s, s'')$
8:     **else**   ▷ AND node
9:         $G_{\text{left}} \leftarrow$ TRAVERSE$(s, s')$
10:       $G_{\text{right}} \leftarrow$ TRAVERSE$(s', s'')$
11:       // BACKUP
12:       $G \leftarrow G_{\text{left}} \cdot G_{\text{right}}$
13:    $G \leftarrow \max(G, v^\pi(s, s''))$  ▷ threshold the return
14:    // UPDATE
15:    $V(s, s'') \leftarrow (V(s, s'')N(s, s'') + G)/(N(s, s'') + 1)$
16:    $N(s, s'') \leftarrow N(s, s'') + 1$
17:    **return** $G$

---

state-spaces $\mathcal{S}$. Instead, we tailor an algorithm for *approximate planning* to the multi-task setting, which we call Divide-and-Conquer MCTS (DC-MCTS). To evaluate $v^\pi$ as sparsely as possible, DC-MCTS critically makes use of two learned search heuristics that transfer knowledge from previously encountered MDPs / tasks to new problem instance: (i) a distribution $p(s'|s, s'')$, called the policy prior, for proposing promising intermediate sub-goals $s'$ for a task $(s, s'')$; and (ii) a learned approximation $v$ to the high-level value $v^*$ for bootstrap evaluation of partial plans. In the following we present DC-MCTS and discuss design choices and training for the two search heuristics.

### 3.1 DIVIDE-AND-CONQUER MONTE CARLO TREE SEARCH

The input to the DC-MCTS planner is an MDP encoding $c_\mathcal{M}$, a task $(s_0, s_\infty)$ as well as a planning budget, i.e. a maximum number $B \in \mathbb{N}$ of $v^\pi$ oracle evaluations. At each stage, DC-MCTS maintains a (partial) AND/OR search tree $\mathcal{T}$ whose root is the OR node $(s_0, s_\infty)$ corresponding to the original task. Every OR node $(s, s'') \in \mathcal{T}$ maintains an estimate $V(s, s'') \approx v^*(s, s'')$ of its high-level value. DC-MCTS searches for a plan by iteratively constructing the search tree $\mathcal{T}$ with TRAVERSE until the budget is exhausted, see Algorithm 1. During each traversal, if a leaf node of $\mathcal{T}$ is reached, it is expanded, followed by a recursive bottom-up backup to update the value estimates $V$ of all OR nodes visited in this traversal. After this search phase, the currently best plan is extracted from $\mathcal{T}$ by EXTRACTPLAN (essentially depth-first traversal, see Algorithm 2 in the Appendix). In the following we briefly describe the main methods of the search. We illustrate DC-MCTS in Figure 1.

**TRAVERSE and SELECT**   $\mathcal{T}$ is traversed from the root $(s_0, s_\infty)$ to find a promising node to expand. At an OR node $(s, s'')$, SELECT chooses one of its children $s' \in \bar{\mathcal{S}}$ to traverse into, including $s = \varnothing$ for not inserting any further sub-goals into this branch. We implemented SELECT by the pUCT

(Rosin, 2011) rule, which consists of picking the next node $s' \in \bar{\mathcal{S}}$ based on maximizing the following score:

$$V(s, s') \cdot V(s', s'') + c \cdot p(s'|s, s'') \cdot \frac{\sqrt{N(s, s'')}}{1 + N(s, s', s'')}, \tag{2}$$

where $N(s, s')$, $N(s, s', s'')$ are the visit counts of the OR node $(s, s')$, AND node $(s, s', s'')$ respectively. The first term is the *exploitation* component, guiding the search to sub-goals that currently look promising, i.e. have high estimated value. The second term is the *exploration* term favoring nodes with low visit counts. Crucially, it is explicitly scaled by the policy prior $p(s'|s, s'')$ to guide exploration. At an AND node $(s, s', s'')$, TRAVERSE traverses into both the left $(s, s')$ and right child $(s', s'')$.[3] As the two sub-problems are solved independently, computation from there on can be carried out in parallel. All nodes visited in a single traversal form a solution tree $\mathcal{T}_\sigma$ with plan $\sigma$.

**EXPAND** If a leaf OR node $(s, s'')$ is reached during the traversal and its depth is smaller than a given maximum depth, it is expanded by evaluating the high- and low-level values $v(s, s''), v^\pi(s, s'')$. The initial value of the node is defined as $\max$ of both values, as by definition $v^* \geq v^\pi$, i.e. further planning should only increase the success probability on a sub-task. We also evaluate the policy prior $p(s'|s, s'')$ for all $s'$, yielding the proposal distribution over sub-goals used in SELECT. Each node expansion costs one unit of budget $B$.

**BACKUP and UPDATE** We define the return $G_\sigma$ of the traversal tree $\mathcal{T}_\sigma$ as follows. Let a refinement $\mathcal{T}_\sigma^+$ of $\mathcal{T}_\sigma$ be a solution tree such that $\mathcal{T}_\sigma \subseteq \mathcal{T}_\sigma^+$, thus representing a plan $\sigma^+$ that has all sub-goals of $\sigma$ with additional inserted sub-goals. $G_\sigma$ is now defined as the value of the objective $L(\sigma^+)$ of the *optimal* refinement of $\mathcal{T}_\sigma$, i.e. it reflects how well one could do on task $(s_0, s_\infty)$ by starting from the plan $\sigma$ and refining it. It can be computed by a simple back-up on the tree $\mathcal{T}_\sigma$ that uses the bootstrap value $v \approx v^*$ at the leafs. As $v^*(s_0, s_\infty) \geq G_\sigma \geq L(\sigma)$ and $G_{\sigma^*} = v^*(s_0, s_\infty)$ for the optimal plan $\sigma^*$, we can use $G_\sigma$ to update the value estimate $V$. Like in other MCTS variants, we employ a running average operation (line 15-16 in TRAVERSE).

## 3.2 DESIGNING AND TRAINING SEARCH HEURISTICS

Search results and experience from previous tasks can improve DC-MCTS on new problems via adapting the search heuristics, i.e. the policy prior $p$ and the approximate value function $v$ as follows.

**Bootstrap Value Function** We parametrize $v(s, s'|c_\mathcal{M}) \approx v^*(s, s'|c_\mathcal{M})$ as a neural network that takes as inputs the current task consisting of $(s, s')$ and the MDP encoding $c_\mathcal{M}$. A straight-forward approach to train $v$ is to regress it towards the non-parametric value estimates $V$ computed by DC-MCTS on previous problem instances. However, initial results indicated that this leads to $v$ being overly optimistic, an observation also made in Kaelbling (1993). We therefore used more conservative training targets, that are computed by backing the low-level values $v^\pi$ up the solution tree $\mathcal{T}_\sigma$ of the plan $\sigma$ return by DC-MCTS. Details can be found in Appendix B.1.

**Policy Prior** Best-first search guided by a policy prior $p$ can be understood as policy improvement of $p$ as described in Silver et al. (2016). Therefore, a straight-forward way of training $p$ is to distill the search results back into into the policy prior, e.g. by behavioral cloning. When applying this to DC-MCTS in our setting, we found empirically that this yielded very slow improvement when starting from an untrained, uniform prior $p$. This is due to plans with non-zero success probability $L > 0$ being very sparse in $\mathcal{S}^*$, equivalent to the sparse reward setting in regular MDPs. To address this issue, we propose to apply Hindsight Experience Replay (HER, Andrychowicz et al. (2017)): Instead of training $p$ exclusively on search results, we additionally execute plans $\sigma$ in the environment and collect the resulting trajectories, i.e. the sequence of visited states, $\tau_{s_0}^{\pi_\sigma} = (s_0, s_1, \ldots, s_T)$. HER then proceeds with *hindsight relabeling*, i.e. taking $\tau_{s_0}^{\pi_\sigma}$ as an approximately optimal plan for the "fictional" task $(s_0, s_T)$ that is likely different from the actual task $(s_0, s_\infty)$. In standard HER, these fictitious expert demonstrations are used for imitation learning of goal-directed policies, thereby circumventing the sparse reward problem. We can apply HER to train $p$ in our setting by extracting any ordered triplet $(s_{t_1}, s_{t_2}, s_{t_3})$ from $\tau_{s_0}^{\pi_\sigma}$ and use it as supervised learning targets for $p$. This is a sensible procedure, as $p$ would then learn to predict optimal sub-goals $s_{t_2}^*$ for sub-tasks $(s_{t_1}^*, s_{t_3}^*)$ under the assumption that the data was generated by an oracle producing optimal plans $\tau_{s_0}^{\pi_\sigma} = \sigma^*$. We have considerable freedom in choosing which triplets to extract from data and use as supervision with HER. In our experiments we use a *temporally balanced* parsing, which creates triplets $(s_t, s_{t+\Delta/2}, s_{t+\Delta})$

---

[3]It is possible to traverse into a single node at the time, we describe several heuristics in Appendix A.3

such that the resulting policy prior should then preferentially propose sub-goals "in the middle" of the task. In Appendix A.4 we discuss this aspect in more detail, and present alternative parsers.

### 3.3 ALGORITHMIC COMPLEXITY OF DC-MCTS

Denoting an optimal plan as $\sigma^*$, the complexity of DC-MCTS with optimal search policy prior $p = p^*$ is $O(|\sigma^*| \cdot |\mathcal{S}|)$. This could potentially be reduced to $O(\log(|\sigma^*|))$ when using progressive widening Coulom (2007); Chaslot et al. for fewer evaluations of $p$ and perfect parallelization of tree traversals across multiple workers; for details see Appendix A.5.

## 4 RELATED WORK

Goal-directed multi-task learning is an important special case of general RL and has been extensively studied. Universal value functions (Schaul et al., 2015) have been established as compact representation for this setting (Kulkarni et al., 2016; Andrychowicz et al., 2017; Ghosh et al., 2018; Dhiman et al., 2018). This allows to use sub-goals as means for planning, as done in several works such as Kaelbling & Lozano-Pérez (2017); Gao et al. (2017); Savinov et al. (2018); Stein et al. (2018); Nasiriany et al. (2019), all of which rely on forward sequential planning. Gabor et al. (2019) use MCTS for traditional sequential planning based on heuristics, sub-goals and macro-actions. Zhang et al. (2018) apply traditional graph planners to find abstract sub-goal sequences. We extend this line of work by showing that the abstraction of sub-goals affords more general search strategies than sequential planning. Work concurrent to ours has independently investigated non-sequential sub-goals planning: Jurgenson et al. (2019) propose a top-down policy gradient approach that learns to predict sub-goals in a hierarchical way. Nasiriany et al. (2019) propose gradient-based search jointly over a fixed number of sub-goals for continuous goal spaces. In contrast, DC-MCTS is able to dynamically determine the complexity of the optimal plan.

The proposed DC-MCTS planner is a MCTS (Browne et al., 2012) variant, inspired by recent advances in best-first or guided search, such as AlphaZero (Silver et al., 2018). It can also be understood as a heuristic, guided version of the classic Floyd-Warshall algorithm which exhaustively computes all shortest paths. In the special case of planar graphs, small sub-goal sets, also known as vertex separators, can be constructed that favourably partition the remaining graph, leading to linear time ASAP algorithms (Henzinger et al., 1997). The heuristic sub-goal proposer $p$ that guides DC-MCTS can be loosely understood as a probabilistic version of a vertex separator. Nowak-Vila et al. (2016) also consider neural networks that mimic divide-and-conquer algorithms similar to the sub-goal proposals used here. However, while we do policy improvement for the proposals using search and HER, the networks in Nowak-Vila et al. (2016) are purely trained by policy gradient methods.

Decomposing tasks into sub-problems has been formalized as pseudo trees (Freuder & Quinn, 1985) and AND/OR graphs (Nilsson, N. J., 1980). The latter have been used especially in the context of optimization (Larrosa et al., 2002; Jégou & Terrioux, 2003; Dechter & Mateescu, 2004; Marinescu & Dechter, 2004). Our approach is related to work on using AND/OR trees for sub-goal ordering in the context of logic inference (Ledeniov & Markovitch, 1998). While DC-MCTS is closely related to the $AO^*$ algorithm (Nilsson, N. J., 1980), which is the generalization of the heuristic $A^*$ search to AND/OR search graphs, interesting differences exist: $AO^*$ assumes a fixed search heuristic, which is required to be lower bound on the cost-to-go. In contrast, we employ learned value functions and policy priors that are not required to be exact bounds. Relaxing this assumption, thereby violating the principle of "optimism in the face of uncertainty", necessitates explicit exploration incentives in the SELECT method. Alternatives for searching AND/OR spaces include proof-number search, recently applied to chemical synthesis planning (Kishimoto et al., 2019). Very recent work concurrent to ours has focused on relevant research directions: Wang et al. (2020) introduce LA-MCTS as a 'meta-algorithm' for black-box optimization, and Chen et al. (2020) propose Retro*, a neural-based A*-like algorithm for molecule synthesis that is also based on AND/OR trees.

## 5 EXPERIMENTS

We evaluate the proposed DC-MCTS algorithm on navigation in grid-world mazes as well as on a challenging continuous control version of the same problem, comparing it to standard sequential MCTS (in sub-goal space) based on the fraction of "solved" mazes by executing their plans. The MCTS baseline was implemented by restricting the DC-MCTS algorithm to only expand the "right"

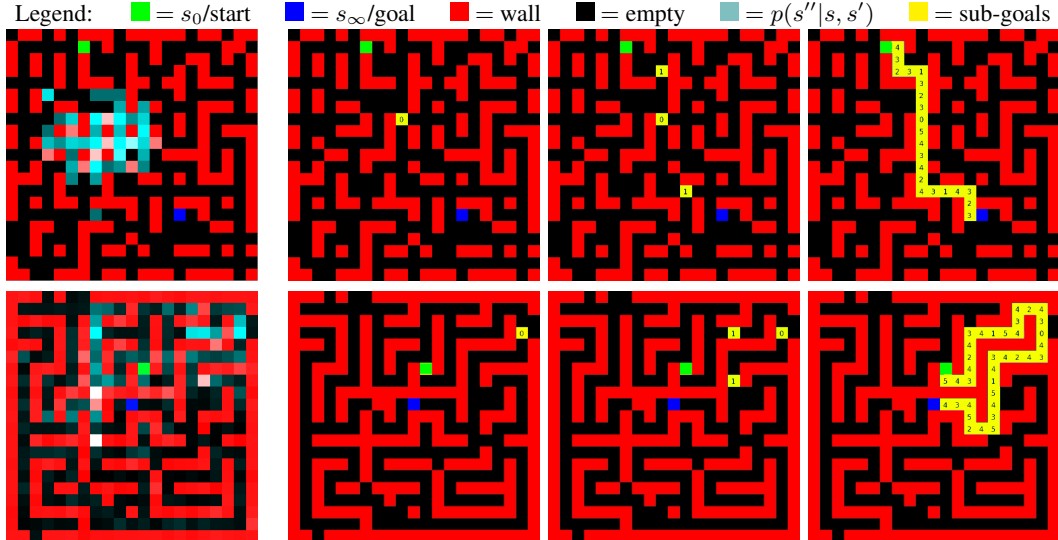

Figure 2: *Left*: Two grid-world maze examples for wall density $d = 0.75$ and $0.95$. In light blue, the distribution over sub-goals induced by the policy prior $p$ that guides the DC-MCTS planner. *Right group*: The first sub-goal, i.e. at depth 0 of the solution tree, approximately splits the problem in half. Next, the *two* sub-goals at depth 1. Last, the final plan with the depth of each sub-goal shown. See supplementary material for full animations.

sub-problem in line 10 of Algorithm 1; the value $G_{\text{left}}$ for the "left" sub-problem is computed as in line 7, i.e. using the low-level value $v^\pi$. This forces MCTS to plan forward and sequentially, as each next step needs to be reachable from the previous state, as evaluated by $v^\pi$. All remaining parameters and design choice were the same for both planners except where explicitly mentioned otherwise.

## 5.1 GRID-WORLD MAZES

Each task consists of a new, procedurally generated maze on a $21 \times 21$ grid with start and goal locations $(s_0, s_\infty) \in \{1, \ldots, 21\}^2$, see Figure 2. Task difficulty was controlled by the density of walls $d$ (under connectedness constraint), where the easiest setting $d = 0.0$ corresponds to no walls and the most difficult one $d = 1.0$ implies so-called perfect or singly-connected mazes. The task embedding $c_\mathcal{M}$ was given as the maze layout and $(s_0, s_\infty)$ encoded together as a feature map of $21 \times 21$ categorical variables with 4 categories each (empty, wall, start and goal location). The underlying MDPs have 5 primitive actions: up, down, left, right and NOOP. For sake of simplicity, we first tested our proposed approach by hard-coding a low-level policy $\pi^0$ as well as its value oracle $v^{\pi^0}$ in the following way. If in state $s$ and conditioned on a goal $s'$, and if $s$ is adjacent to $s'$, $\pi^0_{s'}$ successfully reaches $s'$ with probability 1 in one step, i.e. $v^{\pi^0}(s, s') = 1$; otherwise $v^{\pi^0}(s, s') = 0$. If $\pi^0_{s'}$ is nevertheless executed, the agent moves to a random empty tile adjacent to $s$. Therefore, $\pi^0$ is the "most myopic" goal-directed policy that can still navigate everywhere.

For each maze, MCTS and DC-MCTS were given a search budget of 200 calls to the low-level value oracle $v^{\pi^0}$. We implemented the search heuristics, i.e. policy prior $p$ and high-level value function $v$, as convolutional neural networks (CNNs) which operate on input $c_\mathcal{M}$; details for the network architectures are given in Appendix B.3. With untrained networks, both planners were unable to solve the task (<2% success probability), as shown in Figure 3. This illustrates that a search budget of 200 evaluations of $v^{\pi^0}$ is insufficient for unguided planners to find a feasible path in most mazes. This is consistent with standard exhaustive SSSP / APSP graph planners requiring $21^4 > 10^5 \gg 200$ evaluations for optimal planning in the worst case on these tasks.

Next, we trained both search heuristics $v$ and $p$ as detailed in Section 3.2. In particular, the sub-goal proposal $p$ was also trained on hindsight-relabeled experience data, where for DC-MCTS we used the *temporally balanced parser* and for MCTS the corresponding left-first parser (see Appendix A.4). Training of the heuristics greatly improved the performance of both planners. Figure 3 shows learning

curves for mazes with wall density $d = 0.75$, as mean and std over 20 different hyperparameters. DC-MCTS exhibits substantially improved performance compared to MCTS, and when compared at equal performance levels, DC-MCTS requires 5 to 10-times fewer training episodes than MCTS. An example of a learned sub-goal proposal $p$ for DC-MCTS is visualized in Figure 2 (further examples are given in the Appendix in Figure 8). Probability mass concentrates on promising sub-goals that are far from both start and goal, approximately partitioning the task into equally hard sub-tasks.

Next, we investigated the performance of both MCTS and DC-MCTS in challenging continuous control environments with non-trivial low-level policies. We embedded the grid-world mazes into a physical 3D environment simulated by MuJoCo Todorov et al. (2012), rendering each grid-world cell as 4m×4m cell in physical space. The agent is embodied by a quadruped "ant" body; for illustration see Figure 4. For the low-level policy $\pi^m$, we pre-trained a goal-directed neural network controller that gets as inputs proprioceptive features (e.g. some joint angles and velocities) of the ant body as well as a 3D-vector pointing from its current position to a target

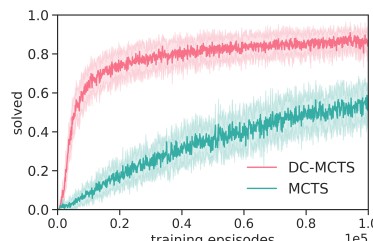

Figure 3: Grid-world mazes.

position. $\pi^m$ was trained to navigate to targets randomly placed less than 1.5 m away in an open area (no walls), using MPO (Abdolmaleki et al., 2018). See Appendix B.4 for more details. If unobstructed, $\pi^m$ can walk in a straight line towards its current goal. However, this policy receives no visual input and thus can only avoid walls when guided with appropriate sub-goals. To establish an interface between the low-level $\pi^m$ and the planners, we used another CNN to approximate the low-level value oracle $v^{\pi^m}(s_0, s_\infty | c_{\mathcal{M}})$: It was trained to predict whether $\pi^m$ will succeed in solving the navigation tasks $(s_0, s_\infty), c_{\mathcal{M}}$. Its input is the corresponding discrete grid-world representation $c_{\mathcal{M}}$ of the maze ($21 \times 21$ feature map of categoricals as described above, details in Appendix). Note that this setting is still challenging: In initial experiments we verified that a model-free baseline (also based on MPO, without HER) with access to state abstraction and low-level controller, only solved about 10% of the mazes after 100 million episodes due to the extremely sparse rewards.

## 5.2 CONTINUOUS CONTROL MAZES

We applied MCTS and DC-MCTS to this problem to find symbolic plans consisting of sub-goals in $\{1, \ldots, 21\}^2$. The high-level heuristics $p$ and $v$ were trained for 65k episodes, exactly as in Section 5.1, except using $v^{\pi^m}$ instead of $v^{\pi^0}$. We again observed that DC-MCTS outperforms by a wide margin the MCTS planner: Figure 5 shows performance of both (with fully trained search heuristics) as a function of the search budget for the most difficult mazes with wall density $d = 1.0$. Performance of DC-MCTS with the MuJoCo low-level controller was comparable to that with the hard-coded low-level policy from the grid-world experiment (with same wall density), showing that the abstraction of planning over low-level sub-goals successfully isolates high-level planning from low-level execution. We did not manage to successfully train the MCTS planner on MuJoCo navigation.

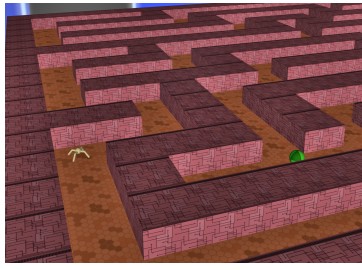

Figure 4: The 'ant', i.e. the agent, should navigate to the green target.

This was likely due to HER, which we found — in ablation studies — essential for training DC-MCTS on both settings and MCTS on the grid-world problem, but not appropriate for MCTS on MuJoCo navigation: Left-first parsing for HER consistently biased the MCTS search prior $p$ to propose next sub-goals too close to the previous sub-goal. This lead the MCTS planner to "micro-manage" the low-level policy, in particular in long corridors that $\pi^m$ can solve by itself. DC-MCTS, by recursively partitioning, found an appropriate length scale of sub-goals, leading to drastically improved performance.

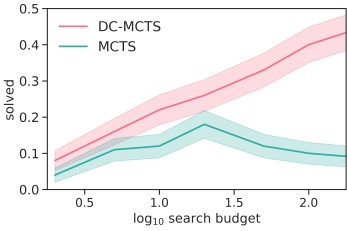

Figure 5: Fraction of solved mazes vs. planning budget.

### 5.3 VISUALIZING MCTS AND DC-MCTS

To further illustrate the difference between DC-MCTS and MCTS planning we can look at an example search tree from each method in Figure 6. Light blue nodes are part of the final plan: note how in the case of DC-MCTS, the plan is distributed across a *sub-tree* within the search tree, while for the standard MCTS the plan is a *chain*. The first 'actionable' sub-goal, i.e. the first sub-goal for the low-level policy, is the left-most leaf in DC-MCTS and the first dark node from the root for MCTS.

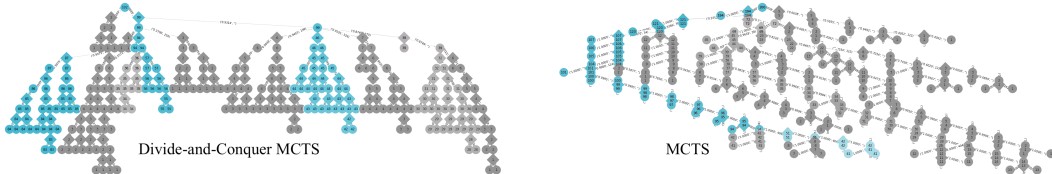

Figure 6: Only colored nodes are part of the final plan: a *sub-tree* for DC-MCTS, a *chain* for MCTS.

## 6 DISCUSSION

To enable guided, divide-and-conquer style planning, we made a few strong assumptions. Sub-goal based planning requires a universal value function oracle of the low-level policy, which often will have to be approximated from data. Overly optimistic approximations can be exploited by the planner, leading to "delusional" plans (Little & Thiébaux, 2007). Joint learning of the high and low-level components can potentially address this issue. In sub-goal planning, at least in its current naive implementation, the "action space" for the planner is the whole state space of the underlying MDPs. Therefore, the search space will have a large branching factor in large state spaces. A solution to this problem likely lies in using learned state abstractions for sub-goal specifications, which is a fundamental open research questions.We also implicitly assumed that low-level skills afforded by the low-level policy need to be "universal", i.e. if there are states that it cannot reach, no amount of high level search will lead to successful planning outcomes.

In spite of these assumptions and open challenges, we showed that non-sequential sub-goal planning has fundamental advantages over the standard approach of search over primitive actions: (i) *Abstraction and dynamic allocation:* Sub-goals automatically support temporal abstraction as the high-level planner does not need to specify the exact time horizon required to achieve a sub-goal. Plans are generated from coarse to fine, and additional planning is dynamically allocated to those parts of the plan that require more compute. (ii) *Closed & open-loop:* The approach combines advantages of both open- and closed loop planning: The closed-loop low-level policies can recover from failures or unexpected transitions in stochastic environments, while at the same time the high-level planner can avoid costly closed-loop planning. (iii) *Long horizon credit assignment:* Sub-goal abstractions open up new algorithmic possibilities for planning — as exemplified by DC-MCTS — that can facilitate credit assignment and therefore reduce planning complexity. (iv) *Parallelization:* Like other divide-and-conquer algorithms, DC-MCTS lends itself to parallel execution by leveraging problem decomposition made explicit by the independence of the "left" and "right" sub-problems of an AND node. (v) *Reuse of cached search:* DC-MCTS is highly amenable to transposition tables, by caching and reusing values for sub-problems solved in other branches of the search tree. (vi) *Generality:* DC-MCTS is strictly more general than both forward and backward goal-directed planning, both of which can be seen as special cases.

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

## A    ADDITIONAL DETAILS FOR DC-MCTS

### A.1    PROOF OF PROPOSITION 1

*Proof.* The performance of $\pi_\sigma$ on the task $(s_0, s_\infty)$ is defined as the probability that its trajectory $\tau_{s_0}^{\pi_\sigma}$ from initial state $s_0$ gets absorbed in the state $s_\infty$, i.e. $P(s_\infty \in \tau_{s_0}^{\pi_\sigma})$. We can bound the latter from below in the following way. Let $\sigma = (\sigma_0, \ldots, \sigma_m)$, with $\sigma_0 = s_0$ and $\sigma_m = s_\infty$. With $(\sigma_0, \ldots, \sigma_i) \subseteq \tau_{s_0}^{\pi_\sigma}$ we denote the event that $\pi_\sigma$ visits all states $\sigma_0, \ldots, \sigma_i$ in order:

$$P((\sigma_0, \ldots, \sigma_i) \subseteq \tau_{s_0}^{\pi_\sigma}) = P\left( \bigwedge_{i'=1}^{i} (\sigma_{i'} \in \tau_{s_0}^{\pi_\sigma}) \wedge (t_{i'-1} < t_{i'}) \right),$$

where $t_i$ is the arrival time of $\pi_\sigma$ at $\sigma_i$, and we define $t_0 = 0$. Obviously, the event $(\sigma_0, \ldots, \sigma_m) \subseteq \tau_{s_0}^{\pi_\sigma}$ is a subset of the event $s_\infty \in \tau_{s_0}^{\pi_\sigma}$, and therefore

$$P((\sigma_0, \ldots, \sigma_m) \subseteq \tau_{s_0}^{\pi_\sigma}) \leq P(s_\infty \in \tau_{s_0}^{\pi_\sigma}). \tag{3}$$

Using the chain rule of probability we can write the lhs as:

$$P((\sigma_0, \ldots, \sigma_m) \subseteq \tau_{s_0}^{\pi_\sigma}) = \prod_{i=1}^{m} P\left( (\sigma_i \in \tau_{s_0}^{\pi_\sigma}) \wedge (t_{i-1} < t_i) \mid (\sigma_0, \ldots, \sigma_{i-i}) \subseteq \tau_{s_0}^{\pi_\sigma} \right).$$

We now use the definition of $\pi_\sigma$: *After* reaching $\sigma_{i-1}$ and *before* reaching $\sigma_i$, $\pi_\sigma$ is defined by just executing $\pi_{\sigma_i}$ starting from the state $\sigma_{i-1}$:

$$P((\sigma_0, \ldots, \sigma_m) \subseteq \tau_{s_0}^{\pi_\sigma}) = \prod_{i=1}^{m} P\left( \sigma_i \in \tau_{\sigma_{i-1}}^{\pi_{\sigma_i}} \mid (\sigma_0, \ldots, \sigma_{i-i}) \subseteq \tau_{s_0}^{\pi_\sigma} \right).$$

We now make use of the fact that the $\sigma_i \in \mathcal{S}$ are *states* of the underlying MDP that make the future independent from the past: Having reached $\sigma_{i-1}$ at $t_{i-1}$, all events from there on (e.g. reaching $\sigma_j$ for $j \geq i$) are independent from all event before $t_{i-1}$. We can therefore write:

$$\begin{aligned} P((\sigma_0, \ldots, \sigma_m) \subseteq \tau_{s_0}^{\pi_\sigma}) &= \prod_{i=1}^{m} P\left( \sigma_i \in \tau_{\sigma_{i-1}}^{\pi_{\sigma_i}} \right) \\ &= \prod_{i=1}^{m} v^\pi (\sigma_{i-1}, \sigma_i). \end{aligned} \tag{4}$$

Putting together equation 3 and equation 4 yields the proposition.    $\square$

### A.2    ADDITIONAL ALGORITHMIC DETAILS

After the search phase, in which DC-MCTS builds the search tree $\mathcal{T}$, it returns its estimate of the best plan $\hat{\sigma}^*$ and the corresponding lower bound $L(\hat{\sigma}^*)$ by calling the EXTRACTPLAN procedure on the root node $(s_0, s_\infty)$. Algorithm 2 gives details on this procedure.

### A.3    DESCENDING INTO ONE NODE AT THE TIME DURING SEARCH

Instead of descending into both nodes during the TRAVERSE step of Algorithm 1, it is possible to choose only one of the two sub-problems to expand further. This can be especially useful if parallel

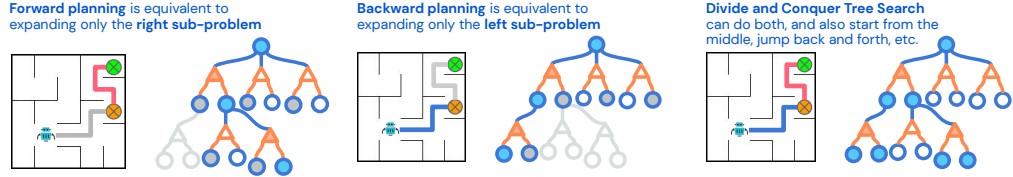

Figure 7: Divide and Conquer Tree Search is strictly more general than both forward and backward search.

---

**Algorithm 2** additional Divide-And-Conquer MCTS procedures

---

Global low-level value oracle $v^\pi$
Global high-level value function $v$
Global policy prior $p$
Global search tree $\mathcal{T}$

1: **procedure** EXTRACTPLAN(OR node $(s, s'')$)
2:    $s' \leftarrow \arg\max_{\hat{s}} V(s, \hat{s}) \cdot V(\hat{s}, s'')$             ▷ choose best sub-goal
3:    **if** $s = \varnothing$ **then**             ▷ no more splitting
4:        **return** $\varnothing, v^\pi(s, s'')$
5:    **else**
6:        $\sigma_l, G_l \leftarrow$ EXTRACTPLAN$(s, s')$         ▷ extract "left" sub-plan
7:        $\sigma_r, G_r \leftarrow$ EXTRACTPLAN$(s', s'')$        ▷ extract "right" sub-plan
8:        **return** $\sigma_l \circ \sigma_r, G_l \cdot G_r$

---

computation is not an option, or if there are specific needs e.g. as illustrated by the following three heuristics. These can be used to decide when to traverse into the left sub-problem $(s, s')$ or the right sub-problem $(s', s'')$. Note that both nodes have a corresponding current estimate for their value $V$, coming either from the bootstrap evaluation of $v$ or further refined from previous traversals.

- *Preferentially descend into the left node* encourages a more accurate evaluation of the near future, which is more relevant to the current choices of the agent. This makes sense when the right node can be further examined later, or there is uncertainty about the future that makes it sub-optimal to design a detailed plan at the moment.

- *Preferentially descend into the node with a lower value*, following the principle that a chain (plan) is only as good as its weakest link (sub-problem). This heuristic effectively greedily optimizes for the overall value of the plan.

- *Use 2-way UCT on the values of the nodes*, which acts similarly to the previous greedy heuristic, but also takes into account the confidence over the value estimates given by the visit counts.

The rest of the algorithm can remain unchanged, and during the BACKUP phase the current value estimate $V$ of the sibling sub-problem can be used.

## A.4 PARSERS FOR HINDSIGHT EXPERIENCE REPLAY

Given a task $(s_0, s_\infty)$, the policy prior $p$ defines a distribution over binary partition trees of the task via recursive application (until the terminal symbol $\varnothing$ closes a branch). A sample $\mathcal{T}_\sigma$ from this distribution implies a plan $\sigma$ as described above; but furthermore it also contains the order in which the task was partitioned. Therefore, $p$ not only implies a distribution over plans, but also a *search order*: Trees with high probability under $p$ will be discovered earlier in the search with DC-MCTS. For generating training targets for supervised training of $p$, we need to *parse* a given sequence $\tau_{s_0}^{\pi_\sigma} = (s_0, s_1, \ldots, s_T)$ into a binary tree. Therefore, when applying HER we are free to choose any deterministic or probabilistic parser that generates a solution tree $\mathcal{T}_{\tau_{s_0}^{\pi_\sigma}}$ from re-labeled HER data $\tau_{s_0}^{\pi_\sigma}$. As mentioned in the main text, the particular choice of HER-parser will shape the search strategy defined by $p$. Possible choices for the parsers include:

1. Left-first parsing creates triplets $(s_t, s_{t+1}, s_T)$. The resulting policy prior will then preferentially propose sub-goals close to the start state, mimicking standard forward planning. Analogously right-first parsing results in approximate backward planning;

2. Temporally balanced parsing creates triplets $(s_t, s_{t+\Delta/2}, s_{t+\Delta})$. The resulting policy prior will then preferentially propose sub-goals "in the middle" of the task. This is the one we used in our experiments;

3. Weight-balanced parsing creates triplets $(s, s', s'')$ such that $v(s, s') \approx v(s's, '')$ or $v^\pi(s, s') \approx v^\pi(s's, '')$. The resulting policy prior will attempt to propose sub-goals such that the resulting sub-tasks are equally difficult.

## A.5 Details on Algorithmic Complexity

Let $c_{v^\pi}$ denote the cost of evaluating the low-level value $v^\pi$ on any sub-problem $(s, s')$. We assume $c_{v^\pi}$ to be independent of $(s, s')$ which holds if e.g. $v^\pi$ is a fixed size neural network. Denote the cost of evaluating the policy prior $p$ on a sub-goal $(s, s', s'')$ with $c_p$. Expanding a new OR node in the search tree incurs a cost of $|S|c_p$ for evaluating $p$ for all children. Assuming the computational cost of tree traversals is negligible, the total cost of running DC-MCTS for $N$ node expansions is thus $N \cdot (|\mathcal{S}|c_p + 2c_{v^\pi})$. The number of expansions to find the optimal (or a sufficiently good) plan strongly depends on the quality of the policy prior (similar to $A^*$ search), making an analysis of the complexity of DC-MCTS challenging for arbitrary $p$. However, if $p = p^*$ is the optimal policy prior – i.e. $p^*(s'|s, s'') = 1$ if $s' \in \sigma^*$ is in the optimal plan $\sigma^*$ for $(s, s'')$ and 0 otherwise – DC-MCTS will construct $\sigma^*$ in the minimal number of step $N = |\sigma^*|$, therefore incurring a cost of $|\sigma^*| \cdot (|\mathcal{S}|c_p + 2c_{v^\pi})$. The dependency on $|\mathcal{S}|$ for the policy prior can be further reduced — in principle down to a constant — using techniques from the literature on MCTS for continuous or large discrete action spaces (e.g. *progressive widening* Coulom (2007); Chaslot et al.). We can compare this to the cost of unguided, standard SSSP planners, which is $|\mathcal{S}|^2 c_{v^\pi}$ as they need to query the low-level value function for all pairs of states $(s, s')$. Therefore, DC-MCTS can be significantly more cost efficient than conventional SSSP planners if a good policy prior can be learned and the cost of evaluating $c_p \lesssim c_{v^\pi}$ is at least comparable to that of evaluating $v^\pi$. Under the assumption $p = p^*$, MCTS and DC-MCTS have the same sample complexity. However MCTS represents the solution as one path of length $|\sigma^*|$ in the search tree, whereas DC-MCTS presents it as a sub-tree with $|\sigma^*|$ nodes. Therefore, if computation can be carried out in parallel (e.g. by batching independent sub-problems at the same level of the sub-tree), the time complexity of DC-MCTS could be drastically reduced compared to MCTS, in the best case (perfect parallelism and balanced binary solution tree) from $\mathcal{O}(|\sigma^*|)$ to $\mathcal{O}(\log(|\sigma^*|))$.

## B Training details

### B.1 Details for training the value function

In order to train the value network $v$, that is used for bootstrapping in DC-MCTS, we can regress it towards targets computed from previous search results or environment experiences. A first obvious option is to use as regression target the Monte Carlo return (i.e. 0 if the goal was reached, and 1 if it was not) from executing the DC-MCTS plans in the environment. This appears to be a sensible target, as the return is an unbiased estimator of the success probability $P(s_\infty \in \tau_{s_0}^{\pi_\sigma})$ of the plan. Although this approach was used in Silver et al. (2016), its downside is that gathering environment experience is often very costly and only yields little information, i.e. one binary variable per episode. Furthermore no other information from the generated search tree $\mathcal{T}$ except for the best plan is used. Therefore, a lot of valuable information might be discarded, in particular in situations where a good sub-plan for a particular sub-problem was found, but the overall plan nevertheless failed.

This shortcoming could be remedied by using as regression targets the non-parametric value estimates $V(s, s'')$ for all OR nodes $(s, s'')$ in the DC-MCTS tree at the end of the search. With this approach, a learning signal could still be obtained from successful sub-plans of an overall failed plan. However, we empirically found in our experiments that this lead to drastically over-optimistic value estimates, for the following reason. By standard policy improvement arguments, regressing toward $V$ leads to a bootstrap value function that converges to $v^*$. In the definition of the optimal value $v^*(s, s'') = \max_{s'} v^*(s, s') \cdot v^*(s', s'')$, we implicitly allow for infinite recursion depth for solving sub-problems. However, in practice, we often used quite shallow trees (depth < 10), so that bootstrapping with approximations of $v^*$ is too optimistic, as this assumes unbounded planning budget. A principled solution for this could be to condition the value function for bootstrapping on the amount of remaining search budget, either in terms of remaining tree depth or node expansions.

Instead of the cumbersome, explicitly resource-aware value function, we found the following to work well. After planning with DC-MCTS, we extract the plan $\hat{\sigma}^*$ with EXTRACTPLAN from the search tree $\mathcal{T}$. As can be seen from Algorithm 2, the procedure computes the return $G_{\hat{\sigma}^*}$ for all OR nodes in the solution tree $\mathcal{T}_{\hat{\sigma}^*}$. For training $v$ we chose these returns $G_{\hat{\sigma}^*}$ for all OR nodes in the solution tree as regression targets. This combines the favourable aspects of both methods described above. In particular, this value estimate contains no bootstrapping and therefore did not lead to

Table 1: Architectures of the neural networks used in the experiment section for the high-level value and prior. For each convolutional layer we report kernel size, number of filters and stride. LN stands for Layer normalization, FC for fully connected,. All convolutions are preceded by a 1 pixel zero padding.

| Value head |
| --- |
| $3 \times 3, 64$, stride = 1 |
| swish, LN |
| $3 \times 3, 64$, stride = 1 |
| swish, LN |
| $3 \times 3, 64$, stride = 1 |
| swish, LN |
| Flatten |
| FC: $N_h = 1$ |
| sigmoid |

| Torso |
| --- |
| $3 \times 3, 64$, stride = 1 |
| swish, LN |
| $3 \times 3, 64$, stride = 1 |
| swish, LN |
| $3 \times 3, 64$, stride = 2 |
| swish, LN |
| $3 \times 3, 64$, stride = 1 |
| swish, LN |
| $3 \times 3, 64$, stride = 1 |
| swish, LN |
| $3 \times 3, 64$, stride = 2 |
| swish, LN |

| Policy head |
| --- |
| $3 \times 3, 64$, stride = 1 |
| swish, LN |
| $3 \times 3, 64$, stride = 1 |
| swish, LN |
| $3 \times 3, 64$, stride = 1 |
| swish, LN |
| Flatten |
| FC: $N_h = $ #classes |
| softmax |

overly-optimistic bootstraps. Furthermore, all successfully solved sub-problems given a learning signal. As regression loss we chose cross-entropy.

## B.2 DETAILS FOR TRAINING THE POLICY PRIOR

The prior network is trained to match the distribution of the values of the AND nodes, also with a cross-entropy loss. Note that we did not use visit counts as targets for the prior network — as done in AlphaGo and AlphaZero for example (Silver et al., 2016; 2018)— since for small search budgets visit counts tend to be noisy and require significant fine-tuning to avoid collapse (Hamrick et al., 2020).

## B.3 NEURAL NETWORKS ARCHITECTURES FOR GRID-WORLD EXPERIMENTS

The shared torso of the prior and value network used in the experiments is a 6-layer CNN with kernels of size 3, 64 filters per layer, Layer Normalization after every convolutional layer, swish (cit) as activation function, zero-padding of 1, and strides [1, 1, 2, 1, 1, 2] to increase the size of the receptive field.

The two heads for the prior and value networks follow the pattern described above, but with three layers only instead of six, and fixed strides of 1. The prior head ends with a linear layer and a softmax, in order to obtain a distribution over sub-goals. The value head ends with a linear layer and a sigmoid that predicts a single value, i.e. the probability of reaching the goal from the start state if we further split the problem into sub-problems.

We did not heavily optimize networks hyper-parameters. After running a random search over hyper-parameters for the fixed architecture described above, the following were chosen to run the experiments in Figure 3. The replay buffer has a maximum size of 2048. The prior and value networks are trained on batches of size 128 as new experiences are collected. Networks are trained using Adam with a learning rate of 1e-3, the boltzmann temperature of the softmax for the prior network set to 0.003. For simplicity, we used HER with the time-based rebalancing (i.e. turning experiences into temporal binary search trees). UCB constants are sampled uniformly between 3 and 7, as these values were observed to give more robust results.

## B.4 Low-level controller training details

For physics-based experiments using MuJoCo (Todorov et al., 2012), we trained a low-level policy first and then trained the planning agent to reuse the low-level motor skills afforded by this body and pretrained policy. The low-level policy, was trained to control the quadruped ("ant") body to go to a randomly placed target in an open area (a "go-to-target" task, essentially the same as the task used to train the humanoid in Merel et al., 2019, which is available at dm_control/locomotion). The task amounts to the environment providing an *instruction* corresponding to a target position that the agent is is rewarded for moving to (i.e, a sparse reward when within a region of the target). When the target is obtained, a new target is generated that is a short distance away (<1.5m). What this means is that a policy trained on this task should be capable of producing short, direct, goal-directed locomotion behaviors in an open field. And at test time, the presence of obstacles will catastrophically confuse the trained low-level policy. The policy architecture, consisting of a shallow MLP for the actor and critic, was trained to solve this task using MPO Abdolmaleki et al. (2018). More specifically, the actor and critic had respectively 2 and 3 hidden layers, 256 units each and elu activation function. The policy was trained to a high level of performance using a distributed, replay-based, off-policy training setup involving 64 actors. In order to reuse the low-level policy in the context of mazes, we can replace the environment-provided instruction with a message sent by a high-level policy (i.e., the planning agent). For the planning agent that interfaces with the low-level policy, the action space of the high-level policy will, by construction, correspond to the instruction to the low-level policy.

## B.5 Pseudocode

We summarize the training procedure for DC-MCTS in the following pseudo-code.

```python
def train_DCMCTS():

    replay_buffer = []

    for episode in n_episodes:
        start, goal = env.reset()
        sub_goals = dc_mcts_plan(start, goal)   # list of sub-goals
        replay_buffer.add(sub_goals)

        state = start
        while episode.not_over() & len(sub_goals) > 0:
            action = low_level_policy(state, sub_goals[0])
            state = env.step(action)
            visited_states.append(state)

            if state == sub_goals[0]:
                sub_goals.pop(0)

        # Rebalance list of visited states to a binary search tree
        bst_states = bst_from_states(visited_states)
        replay_buffer.add(bst_states)   # Hindsight Experience Replay

        if replay_buffer.can_sample():
            neural_nets.train(replay_buffer.sample())
```

## C   MORE SOLVED MAZES

In Figure 8 we show more mazes as solved by the trained Divide and Conquer MCTS.

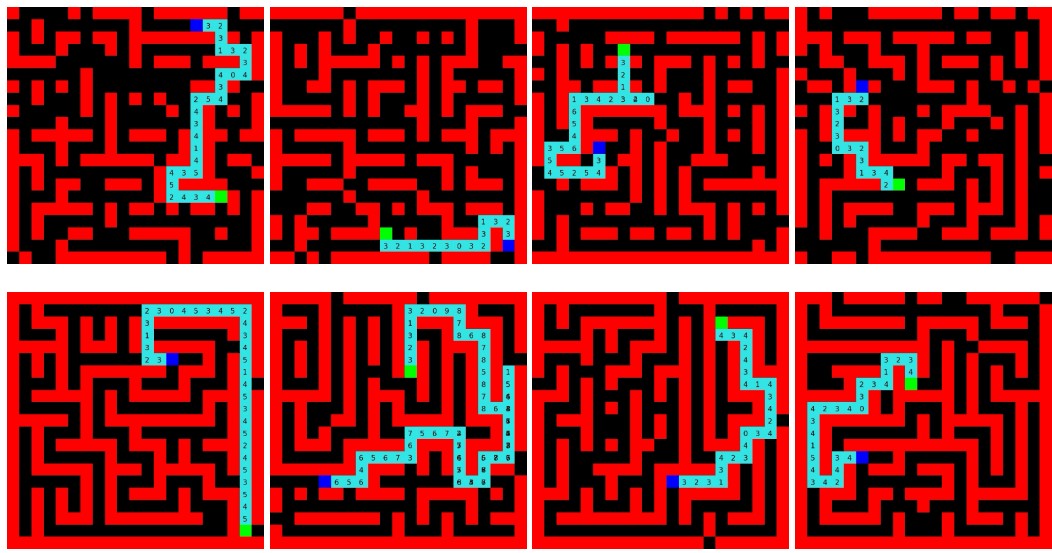

Figure 8: Solved mazes with Divide and Conquer MCTS. ■ = start, ■ = goal, ■ = wall, ■ = walkable. Overlapping numbers are due to the agent back-tracking while refining finer sub-goals.

### C.1   SUPPLEMENTARY MATERIAL AND VIDEOS

Additional material, including videos of several grid-world mazes as solved by the algorithm and of MuJoCo low-level policy solving mazes by following DC-MCTS plans, can be found in the supplementary material.

