# OpenReview forum: "Divide-and-Conquer Monte Carlo Tree Search"
_ICLR.cc/2021/Conference — Reject_

### Official Review · AnonReviewer1 · 2020-10-27
**Solid paper, just some small points where clarification may be useful**

**Rating:** 8
**Confidence:** 4

**Review:**

Summary
---

This paper proposes Divide-and-Conquer Monte Carlo Tree Search (DC-MCTS) for for goal-directed planning problems (i.e. problems where reaching a specific goal state is the objective, like traversing a maze with specified start and goal positions). The assumed setting is one where transition and reward models of the environment are not (necessarily) available, but a low-level goal-directed policy that can attempt to navigate from a given start to a given goal position, as well as value oracle that can return the success probability of the low-level policy on any given task, are available. Planning problems are modelled as AND/OR search trees, where OR nodes are labelled by a start state s and a goal state s'', and AND nodes are labelled by triples (s, s', s''). An OR node has children for every possible state, such that traversing to a child indicates the insertion of the corresponding state as an additional subgoal in between s and s', plus one extra child to indicate the choice of returning the current plan without inserting any additional subgoals. AND nodes have two children; one OR node for the first half (s, s') of the plan, and a second OR node for the second half (s', s'') of the plan. The MCTS can construct a plan by inserting subgoals such that they become easier to solve for the low-level policy by searching this tree.

Strong Points
---

1) Interesting problem setting, way of modelling the problem, and proposed algorithm. Technically sound as far as I can tell.
2) Solid and clear writing.
3) Interesting results.

Weak Points
---

The second paragraph of the Introduction describes two fundamental challenges in sequential planning; (i) assumption of reliable transition model existing, and (ii) credit assignment problem over long time horizons. Then the next paragraph basically starts out (I'm paraphrasing and probably slightly exaggerating here) that we overcome these challenges by changing our assumptions to include that we are already given a goal-directed low-level policy and a value oracle. This may leave the reader wondering "well what if I don't have them?", or "are there practical scenarios where these assumptions are realistic?".

Personally I don't necessarily believe that all research has to immediately have examples of practical applicability readily available, it can still be interesting without it... but in this case, I think it shouldn't be too difficult to actually provide some example situation where these assumptions hold, and including that could make the introduction a bit more convincing. Maybe it's just me, but in particular also the word "oracle" in "value oracle" keeps me scared and wondering for a long time if we're really going to end up needing a hard, ground-truth oracle (unrealistic assumption), only to finally figure out much later in the paper that it's okay for this to just be approximated.

Overall Recommendation
---

I recommend accept. The strong points are clear, and the weak points are quite minor.

Questions for Authors
---

Would it be possible to clarify in more detail how the standard MCTS baseline can be implemented in terms of the Algorithm 1 pseudocode? In Section 5 it's very briefly mentioned that the only changing is restricting DC-MCTS to only expand the "right" sub-problem in line 11 (which I assume should actually refer to lines 9-10). But then G_{left} has no value, and G_{left} is also still required in line 12. Intuitively I think I get the point, that only being allowed to look at the right half of AND nodes forces the algorithm to construct plans again in the same order in which they get executed, and probably by thinking about it more deeply I could figure out how to reconstruct the correct implementation, but either way I feel like it's not super obvious so may be worth clarifying in slightly more detail.

Minor Comments
---

- Final paragraph of Section 1, when listing all the subsequent sections, skips Section 4.
- "where \varnothing is them empty" --> "where \varnothing is the empty" (second paragraph 2.1)
- "MPDs" --> "MDPs" (third paragraph 2.1)
- Even though it's fairly obvious already, I guess I'd prefer "maximizing" rather than "optimizing", to be 100% explicit
- At the end of Section 2: "Figure 5 in Appendix 5.3".... but there is no appendix 5.3 and Figure 5 is not in an appendix.
- "MDP wit a" --> "MDP with a" (first paragraph Section 3)
- "free to chose" --> "free to choose" (A.4)
- "procudre" --> "procedure" (B.5)

---

> ### Author Response · Authors · 2020-11-18
> **Response to R1**
>
> Thanks for the positive outlook on the paper, and for the constructive feedback!
>
> **Maybe it's just me, but the word "oracle" in "value oracle" keeps me scared and wondering for a long time if we're really going to end up needing a hard, ground-truth oracle (unrealistic assumption), only to finally figure out much later in the paper that it's okay for this to just be approximated:** That's a good observation, and it seems to have scared another reviewer too, so thanks for pointing it out. We changed the wording to make it clear that in practice they can be learned (e.g. using HER to pre-train a low-level policy and value function, which is usually very efficient on short goal-directed tasks as it can even be trained with random interaction with the environments).
>
> **Including examples where this would hold could make the introduction a bit more convincing:** We'd be happy to integrate an example. Are you thinking about a high-level example (like going from Paris -> Tokyo), or a specific task in ML?
>
> **Intuitively clear, but clarify in the text how to implement MCTS as DC-MCTS constrained to expanding right node:** We updated this explanation to make it more clear and explicit (also thanks, it was indeed a reference to lines 9-10, and not line 11).
>
> Thank you for pointing out typos and minor comments (even in the appendix!), they are all fixed now.

---

> > ### Comment · AnonReviewer1 · 2020-11-18
> > **Type of example**
> >
> > > We'd be happy to integrate an example. Are you thinking about a high-level example (like going from Paris -> Tokyo), or a specific task in ML?
> >
> > Hmmm not sure that I have a strong preference actually. If you have a specific example ML task in mind that is also easy to explain in one or a handful of sentences, and still easy to digest for the reader, I guess I'd lean towards that. But just a more high-level example may be easier to achieve those goals of being simple to explain and simple to understand, so that might work better.

---

> > > ### Author Response · Authors · 2020-11-21
> > > **Added example to intro**
> > >
> > > We added the following lines to the introduction:
> > >
> > > >Humans can plan efficiently over long horizons to solve complex tasks, such as theorem proving or navigation, and some plans even span over decades (e.g. economic measures): In these situations, planning sequentially in terms of next steps -- such as what arm to move, or what phone call to make -- will cover a tiny proportion of the horizon, neglecting the long uncertainty beyond the last planned step.
> > > The algorithm put forward in this paper is a step in the direction of efficient planners that tackle long horizons by recursively and parallelly splitting them into many smaller and smaller sub-problems.
> > > ----

---

### Official Review · AnonReviewer2 · 2020-10-28
**interesting idea but weak experiments**

**Rating:** 5
**Confidence:** 4

**Review:**

summary:
This paper proposes Divide-and-Conquer Monte Carlo Tree Search (DC-MCTS), a planning algorithm for goal-directed decision-making problems, which makes a plan of the trajectory via recursive hierarchical partitioning. DC-MCTS assumes a (suboptimal) goal-directed low-level policy and its oracle value function. Then, it formulates the given planning problem as finding a sequence of sub-goal states and applies the divide-and-conquer strategy, i.e. split the original task into two sub-tasks (defined as initial state and goal state) and recursively solve them. Unlike the standard MCTS, the decision making of DC-MCTS operates not on the action space but on the state space of the problem, and the decision is made non-sequential way. Experimental results show that DC-MCTS outperforms the MCTS baseline that expands only the right sub-problem.



pros:
- It is interesting to perform planning for sequential-decision making problems in a non-sequential manner via a divide-and-conquer approach.


cons:
- The assumption that goal-conditioned policy and its value is required seems to be too strong, which may limit the generality of the proposed method.

- In the experiments, the baseline algorithm seems to be weak. For example, it would have been nice if it could be compared with other recent methods that combine goal-conditioned RL and planning (e.g. [1], [2]) if applicable. Also, PUCT-like planning algorithms that operate in usual action space could also be a good baseline, where the policy prior can also be trained with HER.



comments and questions:
- It seems to be unfair that the training strategy for the heuristics is different for DC-MCTS and the MCTS baseline. It would be great to see the performance of MCTS that uses the same heuristics training strategy as DC-MCTS.

- The DC-MCTS can be seen as treating 'state space' as 'action space', which implies that it has a much larger branching factor than the traditional action-selecting MCTS since the number of states is typically much larger than the number of actions. Then, why should DC-MCTS be preferred over planning algorithms that operate in usual action spaces? It would be great to see a comparison with the PUCT baseline that works in the original action spaces, where the prior policy and value function is trained using HER.

- (page 7) Did the model-free baseline in the initial experiments use HER? or was it trained only with the actually experienced samples?

- In Figure 2, why the solved ratio did not converge to 1? What kind of behavior can we observe qualitatively when the algorithm fails to solve?

- It seems DC-MCTS is relevant to path planning algorithms. Then, for example, could DC-MCTS be compared with the RRT family?

- Above section 3.2: (line 17-18 in TRAVERSE) -> (line 15-16 in TRAVERSE) ?


[1] Eysenbach et al., Search on the Replay Buffer: Bridging Planning and Reinforcement Learning, NeurIPS 2019
[2] Nasiriany et al., Planning with Goal-Conditioned Policies, NeurIPS 2019

---

> ### Author Response · Authors · 2020-11-18
> **Response to R2 (part 1)**
>
> Thank you for your comments, they helped us clarify several unclear aspects in the paper.
>
> **The assumption that goal-conditioned policy and its value is required seems to be too strong, which may limit the generality of the proposed method.:** DC-MCTS indeed does not provide on its own a low-level goal-directed policy and value function. However, the same assumption is also necessary to use MCTS as a planner in this goal-directed setting without access to an environment model, so this is not a challenge unique to DC-MCTS.
> Moreover, we believe that there has already been great work done in this direction: for example Hindsight Experience Replay (Andrychowicz et al.) typically learns short-term behavior with a goal-directed policy very quickly and can learn from off-policy data and random interactions with the environment.
> Most importantly, consider also that this assumption appears as we replaced another assumption that is typical of planning settings (and is even stronger and unrealistic in most settings, as noted by AnonReviewer1 as well): i.e., having an accurate environment transition model.
> Finally, note that in principle the low-level policy and value functions could be trained at the same time as the planner: Here we focused on the planning part given that it's a novel method in a new setting, but this is an certainly an interesting direction for future work.
>
> **Baseline is weak/RRT:** Please refer to the detailed top-level comment about baselines, and do not hesitate to reach out if you would like to discuss this further.
>
> **PUCT-like planning algorithms that operate in usual action space could also be a good baseline, where the policy prior can also be trained with HER:** This would operate under a different set of assumptions (i.e. access to a transition model), so we do not compare to it in the paper. Nevertheless, when we tried MCTS on a primitive action space (with HER value function) the performance at 25K episodes was 38% solved, vs 71% for DC-MCTS and 14% for MCTS based on sub-goals.
>
> **It seems to be unfair that the training strategy for the heuristics is different for DC-MCTS and the MCTS baseline:** The parser heuristic used for HER in DC-MCTS and MCTS are indeed different: for DC-MCTS we use the "temporally balanced parser" (which essentially restructures a sequence of experiences into a temporally balanced binary search tree), while for MCTS we used the traditional sequential parser as it is used in HER.
> While it would be possible to use the "sequential" parser for DC-MCTS as well -- given that it is strictly more general than MCTS -- it would force it to collapse to forward planning (i.e., it would turn exactly into MCTS), losing its benefit of having a larger space of planning strategies.
> On the other hand, it would not be possible to use the "temporally balanced parser" for MCTS, given that it is constructed to plan sequentially, so the sub-goal predicted at the first level needs to be a reachable sub-goal (and cannot be e.g. one in the middle of the maze).
> Ultimately, the flexibility of DC-MCTS to model different planning strategies (forward, backward, mixed, etc.) is what makes it more general than MCTS. The possibility of using different parsers is a way to tap into this extra flexibility.
> We are happy to clarify this point in the paper, thanks for the question! If you have any further doubts please let us know.
>
> **The DC-MCTS can be seen as treating 'state space' as 'action space'. Why should DC-MCTS be preferred over planning algorithms that operate in usual action spaces?:**
> We believe the answer to your question has two parts:
> - Planning in the action space has its own set of challenges, as it requires a transition model of the world. E.g.: modeling noise and irrelevant dynamics in the observations, inconsequential chaos, and a high sampling rate from the environment which makes long term planning difficult in practice.
> - It depends on the tasks: We posit that there are tasks such as most long-horizon goal-directed planning, where planning in terms of low-level actions leaves too much uncertainty to be covered by the high-level value function. Long horizon plans are difficult to bridge, as even planning hundreds of "next actions" might span a tiny distance that separates the current state from the goal, and essentially forces the value function to approximate a very large horizon.
> Given a good prior over the state-space (and possibly a learned abstraction), for some tasks it might be easier to reason in terms of key-points that have to be reached, instead of low-level actions that need to be performed.
> For example, planning how to go from Paris to Tokyo, you might start by first thinking about how to reach the airport in Tokyo or in Paris. Contrast this with thinking about what low-level action to perform first: taking a step towards the right, or towards the left, or lifting an arm, etc.
>
> (continues)

---

> > ### Author Response · Authors · 2020-11-18
> > **(part 2)**
> >
> > **(page 7) Did the model-free baseline in the initial experiments use HER?** We used standard MPO, so it did not, but it was trained on 100 million episodes (1000x more than DC-MCTS and MCTS). We now mention it explicitly in the paper.
> >
> > **In Figure 2, why the solved ratio did not converge to 1? What kind of behavior can we observe qualitatively when the algorithm fails to solve?** A common failure case occurs when there is an apparently easy solution, which turns out not to work. If the learned prior is very sharp, it might take a budget larger than 200 for the exploration term to compensate.
> >
> > **Above section 3.2: (line 17-18 in TRAVERSE) -> (line 15-16 in TRAVERSE) ?:** You are right, this is fixed now, thanks!

---

### Official Review · AnonReviewer4 · 2020-10-29
**Interesting approach to a well-studied problem but hard to evaluate overall contribution. Needs better experiments.**

**Rating:** 7
**Confidence:** 3

**Review:**

Summary: The paper tackles the problem of multi-task navigation, where the agent sees a different goal-directed (navigation) task at the start of each episode. The paper is reasonably well-written and easy to understand although the text might benefit from an illustrated example. The primary contributions of the paper are algorithmic. The DC-MCTS algorithm seems quite interesting and novel, as far as I can tell, although some of the assumptions (e.g., "task embedding") may reduce applicability. The experiments are a bit thin in their current form which makes it difficult to compare the proposed algorithm to the rather large body of literature in classical planning and robot navigation. With additional empirical analysis, this could be a nice addition to the literature on integrating learning and planning for control in navigation tasks. I look forward to reading the other reviews and the author response.


Detailed Comments
  - The paper tackles the problem of navigation tasks in MDPs, where the a new task is received on every episode (multi-task). The only reward is for reaching the desired goal state. The problem setup assumes that a low-level (naive) goal-directed policy and its value function is available along with a global (optimal) value function. The algorithm leverages this policy in conjunction with a high-level planner to decompose the large navigation task (on which the low-level policy is likely to fail) into smaller sub-tasks. This is done via a sequence of good sub-goal states, recursively broken down roughly in half at each level of the tree. This continues until the sequence of navigation task are "simple" enough for the low-level policy to tackle consecutively. The overall result is a higher probability of successful navigation to the goal compared to a "vanilla" MCTS performing sequential planning.

  - If I've understood correctly, it seems this approach are that the approach does not require a simulator or environment dynamics to construct the search tree. (Right?) Rather, it leverages components (p, \pi, v^\pi) learned or estimated offline from prior experience in different but related navigation tasks. This has the advantage of compressing knowledge about the environment and task into a relatively compact parametric form. However, the experiments suggest that the algorithm is extremely sensitive to noise or error in the learned components. A careful empirical study of the algorithm's sensitivity to the error in these components will likely be required. More on this later.

  - The algorithmic contributions and the intuitions of this paper seem novel to me although I didn't search the classical planning or robotics literature in too much depth. The Related Work section seems relatively complete to me but I'm eager to see if other reviewers can identify relevant related work.

  - Overall, I think the paper could be made easier to understand with a running multi-task example illustrating the core concepts (episode, sequence, plan, trajectory, search tree, solution tree, value functions and policy). Maybe Figure 1 can be expanded and introduced earlier in the paper? Also, it might help to define the given "new MDP" more formally. While this is intuitively clear in the domains considered, I think a more formal description would be better.

  - The main contributions of this paper are algorithmic. Specifically, the proposed algorithm DC-MCTS combines learned components and online planning to efficiently decompose a large search space into sufficiently small ones, so that the given "low-level" components (\pi and v^\pi) can be utilized. The key idea is to decompose the search for a high-probability path between the start and goal states into two (roughly equal "sized") sub-problems by introducing a "sub-goal" (another state) between the original pair as a path constraint. A sequence of such sub-goals book-ended by the start and goal states is a "plan". This seems to be the core idea and is quite simple so I'm surprised it hasn't been explored before in the classical planning literature. As the paper notes, there are strong similarities to classical planning algorithms (AO^*). My own search didn't turn up much so I'd be curious to see what the other reviewers find.

  - I don't think I fully understand the implications of the task embedding (c_M) on the problem setup. Does it imply that DC-MCTS can only tackle fully observable problems? What happens if c_M has error (e.g., learned embedding)? More generally, I'm looking to understand the practicality of the task embedding requirement compared with the more traditional navigation problem setups considered in the deep RL and robotics literature (e.g., https://openreview.net/pdf?id=SJMGPrcle). How would DC-MCTS work if restricted to raw local inputs and prevented from seeing the full maze (via c_M) or received a noisy task embedding?

  - Does DC-MCTS (and the baseline MCTS variant) return a full plan (list of sub-goals) for the given "problem" (state pair at the root) which is then executed by the low-level policy in the environment? I was expecting it to follow the typical anytime planning approach in MCTS where search and behavior are interleaved for each decision in the actual environment. Appendix B.5 suggests that the full plan may be used but maybe B.5 is just an efficient way to get training data for HER and p? Please clarify and consider including these details in the illustration.

  - The experiments are performed in two domains: a grid world and a continuous MuJoCo task. DC-MCTS seems substantially better than the corresponding MCTS in the same search space. The search budget is measured in the number of calls to the oracle value function which allows the two methods to be compared. This section feels a bit under-developed at the moment. The experiments seem to be at a preliminary stage. Although DC-MCTS seems better than the sequential MCTS (in the same search space), the absence of strong baselines on these problems makes these results difficult to interpret. The problem is exacerbated since the baseline seems to be performing very poorly in the MuJoCu domain. Please consider including additional baselines from classical planning / deep RL / robotics (e.g., AO^* variants, Gabor 2019, https://openreview.net/pdf?id=SJMGPrcle). Currently, it's a bit hard to place DC-MCTS in the large body of empirical literature on this subject. Also, when comparing these baselines, I think including additional profiling information (wall-clock runtime, memory usage, etc.) would help the reader get a better understanding of the algorithm's practical performance.

  - The empirical results show that both MCTS algorithms are extremely sensitive to the quality of the global value functions. How sensitive is the overall performance to the choice of the low-level policy? Please consider including additional experiments and analysis of this sensitivity and perhaps additional domains as well. Without these details, it's difficult for the reader to understand the requirements on the learned components and evaluate the algorithm's performance, which is a bit unfortunate as the algorithm is nice. Also, please consider releasing the source code as MCTS experiments can be difficult to reproduce, especially when combined with offline learning setups.

  - Why is the search budget fixed at 200? How do the methods perform with additional computational budget or memory (more nodes expanded at each level)? Would the baseline MCTS perform better given additional resources?

  - Overall, I think the algorithmic ideas and intuition presented in this paper are interesting. The problem setup and DC-MCTS seems to be novel (as far as I can tell) although classical planning and robotics has long investigated similar problems so I could have easily missed something. The implications of the problem setup (e.g., task embedding) are a bit unclear to me. Also, the experimental section is currently a bit thin and doesn't provide much insight into the empirical performance and robustness of the search algorithm against strong baselines and in different computational settings. I look forward to reading the other reviews and the author response.


Minor points
  - n doesn't seem to be defined in Section 2.1.

UPDATE: I thank the authors for their detailed feedback. The authors have addressed a number of concerns and I've increased my score to accept. I continue to think that the empirical section could be easily improved with additional domains, sensitivity to noise, etc. and a careful running example (Fig 1 unfortunately isn't). However, the current draft seems sufficient for publication due to the interesting algorithm (which is novel, as far as I can tell).

---

> ### Author Response · Authors · 2020-11-18
> **Response to R4 (part 1)**
>
> We thank you for the constructive comments. We address these in the following.
>
> **The paper is reasonably well-written and easy to understand although the text might benefit from an illustrated example**: Thanks for the feedback! Based on your suggestion we added a new figure in the paper (Figure 1, page 1) to illustrate DC-MCTS.
>
> **Model:** You are correct in that DC-MCTS does not require an environment transition and reward model. Planning is done based solely on low-level policy and value function.
>
> **Error in learned components:** The learned components in DC-MCTS are the high-level search policy and value function which guide the search. It is correct that empirical performance of DC-MCTS depends strongly on these components (e.g. as can be seen during learning Fig 2), but this is analogous to e.g. A* search -- in both cases search efficiency can be massively increased over e.g. BFS by appropriate search heuristics. However, basic MCTS theory still applies to DC-MCTS guaranteeing that due to exploration bonus, the optimal plan will be found in the limit of infinite search budget irrespective of the learned components. Furthermore, as the components are indeed learned (making use of casting MCTS as policy improvement operator; see e.g. Silver et al.: Mastering the game of go without human knowledge), noise or imperfections in high-level policy and value functions decrease with repeated planning as shown in Fig. 2.
>
> **Choice of low-level policy**: The low-level policy and value function in DC-MCTS play roles analogous to that of the environment model in regular planners. Regarding the latter: It is notoriously difficult to reason about the impact of model imperfections on planning quality (e.g. see [Talvite. Agnostic System Identification for Monte Carlo Planning] and references therein); to the best of our knowledge, this has led to de facto standard in the literature of treating questions of model accuracy and planning efficiency orthogonally. Similarly, we found it difficult to theoretically characterize planning error in terms of low-level component error.
>
> **How sensitive is the overall performance to the choice of the low-level policy?:** We tested this in the context of the first experiment by changing the low-level policy to a more powerful one. Instead of reliably walking 1 step in every direction (adjacent cells -> most myopic low-level policy), we made it such that it could walk up to 3 steps. This means that in practice the effective horizon of a successful plan is shortened by a factor 3 (and therefore the problem is easier to plan). At 50K episodes DC-MCTS reached 83% accuracy, vs 51% for MCTS.
> In practice we observe that for longer horizons (i.e. weaker low-level policy) the gap between DC-MCTS and MCTS increases, which is consistent with the expectations for a divide-and-conquer approach.
>
> **Any-time algorithm / replanning:** There might be a slight confusion about the concepts of anytime planning and re-planning. DC-MCTS is an any-time algorithm: For any given search budget it will return the full plan (consisting of the full sequence of sub-goals) -- i.e. it can be interrupted at any time and will return the best plan so far. An orthogonal issue is that of replanning: In our experiments, we do not replan, i.e. we execute the entire plan feeding the first sub-goal to the low-level policy, switching to the next sub-goal the previous is reached. It is straight-forward to add a re-planning step into this procedure at any moment, e.g. to plan from scratch after every low-level action executed by the low-level policy. In practice, re-planning is essential to robustify against planning errors, but as we expect all planning methods to gain from replanning and for the sake of simplicity we only compared MCTS and DC-MCTS without replanning.
>
> **Task embedding / family of MDPs / partial observability:** In the new version of the paper, we clarified these points. We summarize and clarify in the following. The task embedding is an arbitrary statistic of the concrete MDP of each episode, it can e.g. be empty or noisy. Non-trivial task embeddings can lead to better planning performance by providing the low- and high-level components with additional information. In the maze experiments, it was chosen as the maze layout. We do indeed require that each task is given by an MDP, i.e. currently DC-MCTS cannot handle partially observed domains (POMDPs). In general, planning in POMDPs is extremely challenging, as it requires strong models of the uncertainty in the domain which are very difficult to learn. Nevertheless, we are currently considering extending DC-MCTS to POMDPs using techniques such as [Silver, Veness. Monte-Carlo Planning in Large POMDPs].

---

> > ### Author Response · Authors · 2020-11-18
> > **(part 2)**
> >
> > **Varying performance with search budget:** In initial experiments, we found that a search budget of 200 was a good balance between computational cost and fast learning. An example for search performance for varying search budgets is shown in Figure 5 (previously 4) of the manuscript.
> >
> > **Insight into empirical performance of the method:** We added several visualizations of the trained DC-MCTS into the supplementary material zip. These show that empirically DC-MCTS does indeed converge to a strategy of divide-and-conquer, which roughly splits sub-goals in half at each iteration and solves in parallel different sub-problems.
> >
> > We hope we could address your concerns. Thanks for helping us improve the paper!

---

### Official Review · AnonReviewer3 · 2020-10-29
**Borderline paper**

**Rating:** 5
**Confidence:** 3

**Review:**

The authors propose an novel planning algorithm that treats each iteration in the divide-and-conquer algorithm as an MDP decision step and perform Monte-Carlo tree search (MCTS) on top of it. Experiments on both discrete and continuous planning task demonstrate the effectiveness of the proposed algorithm over existing MCTS planners.

The idea is interesting and novel in the sense that for existing methods in planning, in each step we expand the search tree outward by a step size, but not split the search space in half and work on each subproblem. From this aspect, two highly related papers are worth discussing/comparing to in the paper in order to better position the current work. In LA-MCTS\cite{wang2020learning}, each node also represents a split of the search space. The difference is that their work is on black-box optimization and they return a single point instead of a plan. For goal-directed planning, spliting the original problem into subproblems create an AND/OR tree search space. To tackle this, Retro*\cite{chen2020retro} proposes a neural version of the A* algorithm on AND/OR trees (AO* algorithm). Although they are tasking a different problem, their retrosynthesis task can also be seen as a goal-based problem where the goal is the molecule they want to synthesize.

The methodology is presented with good writing and organization. I am able to understand the approach clearly but with the following question. If (at the beginning) the policy prior is not trained well, it is possible that the prior keep proposing subgoals that are on the same plan which is suboptimal. Therefore many samples will be wasted. Is the exploration mechanism in PUCT able to handle this? Or the PUCT is going to `explore' (be trapped) inside the same plan again and again and not able to propose an near-optimal plan.

To evaluate the proposed algorithm, the authors conduct experiments on both discrete and continuous grid worlds and compare the performance with MCTS. The analysis of both experiments are comprehensive. However, my concern on this part is that the evluation is not convincing since there is only one baseline which is MCTS, which itself is not the best planner for goal-directed planning. For planning baselines, consider adding 2D A* (discrete case) or RRT* \cite{karaman2011sampling} (continuous case). Or the authors can compare with hierarchical RL methods. To improve the paper (but might not neccessary), I suggest the authors to replace the continuous grid world with other control tasks such as robot arm control. Currently the two experiments are too alike.

In general, I think the paper presents an novel approach which is promising and potentially beneficial to other areas in machine learning. However the current paper is not ready for publication in its current form due to the reasons listed above. I would consider raising my score if the authors address my concerns in the revision.


@article{wang2020learning,
  title={Learning Search Space Partition for Black-box Optimization using Monte Carlo Tree Search},
  author={Wang, Linnan and Fonseca, Rodrigo and Tian, Yuandong},
  journal={arXiv preprint arXiv:2007.00708},
  year={2020}
}


@inproceedings{chen2020retro,
  title={Retro*: Learning Retrosynthetic Planning with Neural Guided A* Search},
  author={Chen, Binghong and Li, Chengtao and Dai, Hanjun and Song, Le},
  booktitle={The 37th International Conference on Machine Learning (ICML 2020)},
  year={2020}
}

@article{karaman2011sampling,
  title={Sampling-based algorithms for optimal motion planning},
  author={Karaman, Sertac and Frazzoli, Emilio},
  journal={The international journal of robotics research},
  volume={30},
  number={7},
  pages={846--894},
  year={2011},
  publisher={Sage Publications Sage UK: London, England}
}

Update after rebuttal:
Thanks for the response! The authors resolve my concerns except for the baselines. I agree that MCTS is the most relevant baseline, but the authors should also include stronger baselines as well. Obviously, MCTS is far from the best performing algorithm on grid world navigation tasks. I encourage the authors to either adding stronger baselines or switch to another task where MCTS is the dominant algorithm. I still think the current evaluations are not adequate to publish in a top-tier conference. Therefore I will keep my score unchanged.

---

> ### Author Response · Authors · 2020-11-18
> **Response to R3**
>
> Thank you for the constructive feedback!
>
> **Wang (2020) and Chen (2020):** thanks for the very recent references, we worked both of these relevant concurrent works into the related work section.
>
> **Would pUCT get trapped with a poor policy prior (e.g. at the beginning)?:** that's a good question, in practice we observed the following:
> 1) The policy prior at the beginning of training is bad, and as such it leads to bad plans: it proposes almost exclusively plans that are completely unachievable. We hint at this in the section on the policy prior, but basically this results in extremely slow progress.
> 2) MCTS is a policy improvement operator, but it could still get stuck if the policy is poorly initialized (which is what happens in practice with a random network initialization.
> 3) The added exploration bonus guarantees that it will eventually escape.
>
> Additionally, we found that HER significantly speeds up this process, which is consistent with the observations made in the original paper on HER regarding goal-directed RL.
> In order to apply it to DC-MCTS, we had to find a way to adapt HER to our divide-and-conquer setting, since in its original implementation it would "push" the prior towards sub-goals that are directly reachable (collapsing DC-MCTS to MCTS). We ended up with the "binary search tree" parser, if you are interested we describe a couple more parses for HER in appendix A.4.
>
> **Analysis of both experiments is comprehensive, but MCTS is not the best planner for goal-directed RL:** Please refer to the top-level comment about baselines and do not hesitate to reach out if you would like to discuss this further.

---

### Author Response · Authors · 2020-11-18
**To all reviewers**

We thank you all for their constructive feedback and are glad you found the method to be interesting and novel.
There have been concerns about the lack of appropriate, “strong” baselines from prior literature for the experiments. We address this concern in detail in the following, but in summary, we want to clarify that 1) we are interested in general goal-directed planning, not in navigation per se, 2) that the MCTS baseline, essentially AlphaZero, is appropriate and very strong and 3) that for the novel setting considered (given low-level policy and value function) there are no standard baselines in the literature to the best of our knowledge.:

1) The method we propose is a general goal-directed planning method, and not meant only for navigation per se. The focus of this paper is on the algorithmic side.
2) The MCTS baseline is essentially equivalent to the core method of AlphaZero (Silver et al, Mastering Chess and Shogi by Self-Play with a General Reinforcement Learning Algorithm): a learned policy network and value network are trained using MCTS as a policy improvement operator. Besides the strong performance obtained on several challenging tasks, the method is also theoretically grounded, in the way it finds the optimal plan in the limit of infinite search budget (irrespective of the learned components). Moreover, unlike in Silver et al, due to the goal-directed nature of the setting we investigate, we augment its training process with Hindsight Experience Replay. Overall, this version of MCTS we adopted is a strong baseline.
3) We maintain that MCTS is the most relevant baseline, as DC-MCTS is a direct extension of it. The two share both the conceptual backbone, the theoretical guarantees, and even the code: DC-MCTS is a generalization of MCTS, and as such MCTS is an ablation that ensures that we compare similar implementations (guaranteeing the soundness of the results) and it allows us to investigate how DC-MCTS benefits from dropping the sequentiality constraint.
Finally, we briefly discuss why we see the other baselines mentioned as not applicable:
- AO* [Nilsson, N. J. Principles of AI]: This method requires, just like A*, a valid search heuristic which is a bound on the value. Such a valid heuristic is not given in the setting we consider, therefore AO* is not applicable. AlphaZero (=MCTS with learned heuristic, as applied in our experiments) is a natural extension of this: The exploration bonus prevents premature discard of potential sub-goals.
- [Mirowski et al. Learning to Navigate Complex Mazes.] and other model-free RL baselines: These methods seem to apply to similar RL environments, but address very different fundamental problems: RL vs Planning. While we address the latter in the paper (known low-level policy and value function), the former does not assume anything. It is difficult to see how fair comparisons could be made between these, as they assume drastically different states of knowledge initially. We did compare DC-MCTS with a SOTA model-free method (MPO, sec 5.2), finding (unsurprisingly) that it fails to learn the task.
- [Eysenbach et al., Search on the Replay Buffer: Bridging Planning and RL] assumes access to a non-parametric memory, i.e. observed transitions, for each new MDP (replay buffer) and is therefore not comparable. Instead, we could use DC-MCTS to replace Dijkstra’s search algorithm in their approach to change how the replay buffer is searched. As explained in the paper, Dijkstra’s algorithm requires a much larger search budget to obtain comparable results to DC-MCTS.
- [Nasiriany et al., Planning with Goal-Conditioned Policies]: This is concurrent work in a similar setting, with one essential difference: Their proposed setting assumes continuous states, whereas we consider discrete states. Analogously to the fact that methods in continuous control and tabular RL or continuous vs discrete optimization are often not comparable, DC-MCTS and the work of Nasiriany et al address orthogonal problems.
- RRT [Karaman et al. Sampling-based Algorithms for Optimal Motion Planning]: RRT and related algorithm are explicitly designed to solve only navigation tasks and make corresponding strong assumptions, e.g. that the states of the MDPs are embedded in Euclidean space -- whereas DC-MCTS is designed for goal-directed planning in general MDPs. Vanilla RRT randomly adds states to evaluate, and as such it is unlikely to outperform fully trained DC-MCTS even on the presented navigation task as it is, and should be no more efficient than untrained DC-MCTS with a uniform search prior. A neural RRT with learned value functions e.g. [Chen et al. Learning to Plan in High Dimensions via Neural Exploration-Exploitation Trees] should perform much better. Again, the latter makes strong navigation-specific assumptions and would not be applicable to solve general planning tasks.

Based on your recommendations, we added a figure to illustrate DC-MCTS (Figure 1) and clarified several aspects.

---

### Decision · Program_Chairs · 2021-01-07
**Final Decision**

**Decision:**

Reject

**Comment:**

This paper proposes an MCTS approach to goal-conditioned planning, where the search generates high-level sequences of subgoals for low-level policies. This top-level planner is basically a search-based implementation of SSST for potential gain in computational requirements with the help from the advanced search techniques behind PUCT that combines MCTS and prior information.

Reviewers generally agreed that this is an interesting and novel approach to planning and reinforcement learning. However, reviewers generally expressed that the experiments fall short to convince readers that this technique has greater impact and potential for a wider range of applications, other than GridWorld-like environments. Authors are encouraged to provide a more extensive set of experiments, adding more variety to the domains and ablation studies such as the impact of incorrect prior on the overall search performance.